# ProtoX: Explaining a Reinforcement Learning Agent via Prototyping

**Ronilo J. Ragodos**
Department of Business Analytics
University of Iowa
Iowa City, IA 52242
ronilo-ragodos@uiowa.edu

**Tong Wang** *
Department of Business Analytics
University of Iowa
Iowa City, IA 52242
tong-wang@uiowa.edu

**Qihang Lin**
Department of Business Analytics
University of Iowa
Iowa City, IA 52242
qihang-lin@uiowa.edu

**Xun Zhou**
Department of Business Analytics
University of Iowa
Iowa City, IA 52242
xun-zhou@uiowa.edu

## Abstract

While deep reinforcement learning has proven to be successful in solving control tasks, the "black-box" nature of an agent has received increasing concerns. We propose a prototype-based post-hoc *policy explainer*, ProtoX, that explains a black-box agent by prototyping the agent's behaviors into scenarios, each represented by a prototypical state. When learning prototypes, ProtoX considers both visual similarity and scenario similarity. The latter is unique to the reinforcement learning context, since it explains why the same action is taken in visually different states. To teach ProtoX about visual similarity, we pre-train an encoder using contrastive learning via self-supervised learning to recognize states as similar if they occur close together in time and receive the same action from the black-box agent. We then add an isometry layer to allow ProtoX to adapt scenario similarity to the downstream task. ProtoX is trained via imitation learning using behavior cloning, and thus requires no access to the environment or agent. In addition to explanation fidelity, we design different prototype shaping terms in the objective function to encourage better interpretability. We conduct various experiments to test ProtoX. Results show that ProtoX achieved high fidelity to the original black-box agent while providing meaningful and understandable explanations.

## 1 Introduction

Deep reinforcement learning (DRL) has been widely used to solve various control problems such as autonomous driving [1], gaming [2], etc. Despite superior performance, DRL agents are often criticized for being "black-boxes" in nature and unable to provide human-understandable explanations for their actions. Popular solutions to interpreting a black-box agent include attention-based methods [3] and saliency maps [4, 5, 6]. These explanations, however, only inform users of where the agent "looks", but do not provide a more specific explanation for *why an agent takes an action in a given state*. Some recent methods answer this question by formulating a classification problem and building a decision tree from state information to predict an action [7]. However, tree-based models face the limitation of feature representation, that they can only work with meaningful features in order to

---

*corresponding author

36th Conference on Neural Information Processing Systems (NeurIPS 2022).

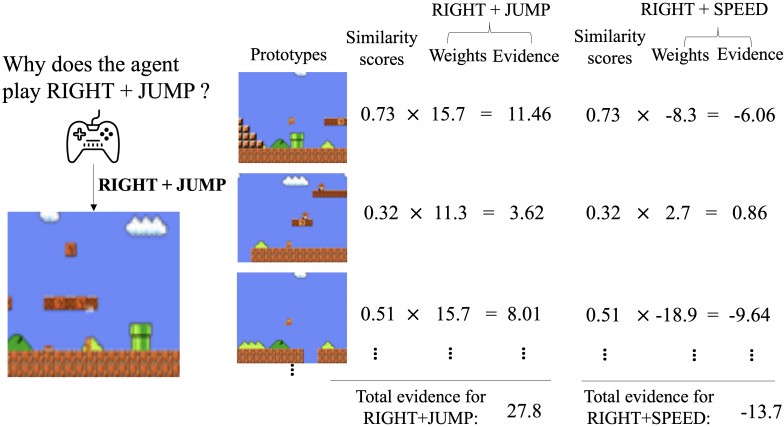

Figure 1: Given an input state where Mario stands in front of a pipe and the agent plays the action RIGHT + JUMP, ProtoX explains this action by collecting *evidence* from the similarities to the prototypical behaviors. The evidence is the sum of similarities between the input state and the prototypes, weighted by the action connections. An action connection represents how strongly a prototype supports the prediction of an action and in which direction. Thus, the action with the largest evidence (RIGHT + JUMP with evidence 27.8) is predicted for the given state.

make sense to humans. They do not naturally apply to unstructured data such as images and texts unless human-understandable information is manually extracted.

This paper aims to *explain* a black-box agent that works with unstructured data. To make the explanations understandable, we make use of the concept of *prototypes* from recent DNN models for image classification [8, 9] and text classification [10]. A prototype is an instance (e.g. an image) from data that is representative of a prominent feature for a certain class. For example, the red-bellied woodpecker class could have a prototype for its red head and a prototype for its wings with a zebra-like pattern, etc. Then, when there is a new input, the input can be compared against all prototypes trained from data. If the input also has a red head and zebra-like pattern on the wings, the weighted sum of similarities in the red-bellied woodpecker class will be high, and thus the input will be predicted to be in this class.

In this paper, we propose a policy explainer called ProtoX, which explains an action by relating an input state to a set of prototypical behaviors learned from the demonstrations from the agent. Figure 1 shows an example of ProtoX explanation for an agent playing the Super Mario game. ProtoX learns a set of prototypical states during training, which are most representative of the agent's behavior.

In summary, ProtoX prototypes an agent's behavior into a small set of representative behavior patterns, each represented by a prototype, and explains the agent's action based on these basic patterns. Therefore, a key determinant of the success of ProtoX is the prototypes. An important consideration when computing the similarity between an input and a prototype is the **visual** similarity. Our definition of similarity must appropriately balance the aspects of visual and contextual similarity. Otherwise, e.g. if two states look entirely different and are assigned high similarity, human users may get confused and discredit the explanation. In ProtoX, we design a self-supervised training step for the encoder to learn visual similarity. Usually, images that are closer to each other in time and receive the same actions are more visually similar than those far away or receiving different actions. Thus, we create a siamese net [11, 12] that utilizes a modified form of the time-contrastive loss. Our pre-training routine yields encoders whose feature spaces cluster together visually similar states which occur closely in time and have the same corresponding actions.

In the supervised learning setting, visual similarity would be the only consideration, given the nature of image classification tasks. However, in a reinforcement learning setting, two visually similar states could receive different actions (such as the flip points we define later), and two visually dissimilar states could represent a similar "scenario" that requires the same action. Therefore, the similarities also need to take into account **scenario** similarity. Here, a scenario represents a collection of states in which an agent performed the same action for similar reasons. For example, in Figure 1, an action RIGHT+JUMP has a high similarity score to prototypes that represent different scenarios for

jumping, which could be that Mario encounters a pipe (the 1st prototype), a hole (the 3rd prototype), or preparing to hit a question mark brick (the 2nd prototype). In fact, the scenario similarity brings more value to the explanation, since it helps users to think about what is common between two visually dissimilar states that leads to the same action. To allow the feature representations to incorporate scenario similarity, we add a near-isometry layer after the encoder, which applies a linear transformation of the output of the encoder to allow some states to move closer to each other.

ProtoX is trained end-to-end via imitation learning, by learning from the demonstrations of the black-box agent to be explained. We choose behavioral cloning [13] in this paper because it requires minimal information from both the agent and the environment: only demonstrations from the agent and no access to the environment or the agent. Therefore, it is most applicable in practice. Many other imitation methods, such as GAIL-based methods and inverse reinforcement learning methods, require more information from the environment or/and the agent, such as knowing the transition functions, executing a policy during training, etc.

We summarize the contributions of ProtoX. First, ProtoX is *the first to explain the actions of a reinforcement learning agent via prototypes*. Most importantly, the reinforcement learning agent is used as a black-box, which means ProtoX only needs to learn from demonstrations and does not need to access the internals of the agent, which makes it easier to use in practice. Meanwhile, ProtoX also does not need to access or interact with the environment. Second, we design an encoder to preserve visual similarity for better interpretability via self-supervised learning and an isometry layer. It allows ProtoX to adapt to both visual and contextual notions of similarity. Third, while the existing interpretable baseline (VIPER) only works well on structured data, ProtoX can more naturally work with unstructured data.

## 2 Related Work

RL interpretation can be categorized into two main groups: inherently interpretable RL models and post-hoc explanation approaches. Inherently interpretable RL models generate policies that are intrinsically understandable, such as Neurally Directed Program Search [14], fuzzy RL policies [15], hierarchical policies [16, 17, 18], or extract time step importance within episodes as strategy-level explanations [19]. Post-hoc methods, like the proposed ProtoX, aim to explain a given black-box agent to help users understand the agent's decision-making process towards an action. The goal is not good task performance but high fidelity in explaining a given black-box agent, regardless of the own performance of the agent.

Based on the type of data they work with, post-hoc methods can also be split into two areas. The first class of methods only work with structured data, such as genetic programming [20], VIPER[7] and other rule-based methods [21, 22, 23], expected consequences [24], causal lens [25], complementary RL [26], etc. These methods often adopt existing forms of interpretable models from supervised learning, such as rules [7, 26]. However, since the explanations need to directly use state features, they do not naturally work with unstructured data. To learn video games from pixels, they require meaningful features to be extracted *manually*. The other type of post-hoc methods work with structured data. Examples include saliency-based methods [27, 28, 29, 26, 4], reward decomposition [30], and interestingness elements [31]. The explanations, however, are either only at the global level or too abstract, such as saliency maps that only inform users of where the agent "looks" but do not directly explain the policy.

Another stream of work that aims to provide some interpretability stems from the family of imitation learning. For example, inverse reinforcement learning (IRL) methods [32, 33, 34] aim at learning a reward function under which the observed demonstrations of an agent is near-optimal. The interpretability, however, is only achieved through the learned reward function with a pre-defined structure (e.g., linear combination of state features) rather than understanding the policy itself. Methods based on generative adversarial imitation learning (GAIL) [35] such as xGAIL [36] and infoGAIL [37] provide indirect explanations to the learned policy in terms of global and local feature importance or latent factors that govern the policy generation. However, none of them provide a direct explanation of the expert's actions under each state that are intuitive to humans. More importantly, all of the above methods require intensive interactions with the environment or the agent during training (such as executing a policy), which are often not accessible in most practical situations. On the other hand, our proposed ProtoX model is completely offline and requires no environment or

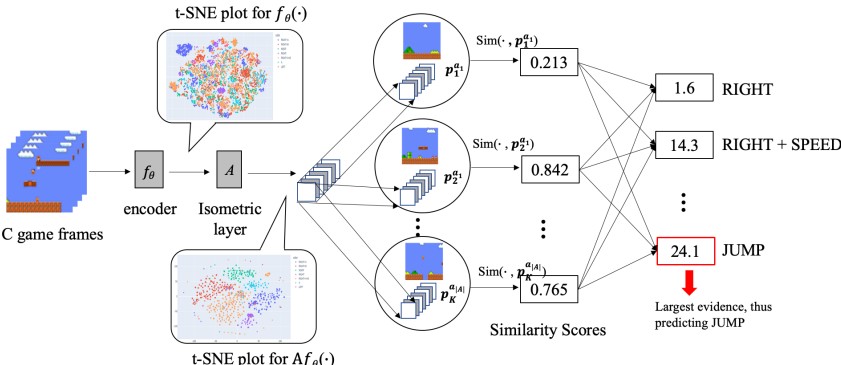

Figure 2: The model architecture of ProtoX.

agent interaction. It also needs no additional information from the environment such as the transition function, reward function, etc. Another work that is relevant to ProtoX is VINN [38], which also learns from visual demonstrations.VINN uses BYOL [39] to train a self-supervised encoder for visual representation and then uses KNN based regression for action prediction. Since VINN was not designed as an explainer, it does not consider the visual similarity and the human-understandable explanation as ProtoX does.

## 3 Prototype-based Policy Explainer

We consider discrete-time partially observable Markov decision processes (POMDP), which include video games such as the OpenAI Gym classic control games[40]. We denote the observation (e.g., frame of video game) at time $t$ by $o_t$ and the $C$ latest observations at time $t$ by $s_t = (o_t, o_{t-1}, \ldots, o_{t-C+1})$. We call $s_t$ the state at time $t$ and denote the set of all states across all times as a state space $\mathcal{S}$.[2] Let $\mathcal{A}$ be the action space of the POMDP, and consider a policy that chooses an action $a \in \mathcal{A}$ at time $t$ based on the $C$ latest observations, namely, a mapping $\pi : \mathcal{S} \to \mathcal{A}$.

Given a policy by an expert $\pi^* : \mathcal{S} \to \mathcal{A}$, our goal is to build a prototype-based policy explainer that is parameterized by $\phi$, denoted as $\pi_\phi : \mathcal{S} \to \mathcal{A}$, called ProtoX, such that $\pi_\phi(s) = \pi^*(s), \forall s$. We train $\pi_\phi$ via behavioral cloning, which proceeds with two steps. First, we apply $\pi^*$ to the POMDP to generate a set of state-action trajectories and decompose all trajectories into a dataset of the expert's actions at different states, denoted by $\mathcal{D} = \{(s_i, \pi^*(s_i))_i\}$. Next, we train a classifier to predict actions from states.

We present the architecture of ProtoX, which consists of four main components, an encoder, a near-isometry layer, a prototype layer, and a fully connected layer. See Figure 2 for an illustration of the architecture. Below, we describe its main components and then formulate the learning objective, which considers both the fidelity and the interpretability.

### 3.1 Feature Representations Preserving Visual and Scenario Similarity

A key determinant of the success of ProtoX is that the feature representation of the states that can respect the visual similarity *and* reflect scenario similarity. Therefore, ProtoX needs to recognize task-relevant features from the images to benefit the downstream task, as well as understand what is considered "visually similar". However, we do not have labeled data that supervises an encoder to determine which images are similar and which are not. Therefore, we use a self-supervised learning approach. We construct an objective via contrastive learning and train an encoder $f_\theta$ parameterized by $\theta$ using a siamese variational auto-encoder (VAE) [41]. The contrastive portion of our objective is similar to the time-contrastive triplet loss from [11], based on the idea that images that are closer to each other in time are more likely to be visually similar to each other. Our loss function differs slightly in that it uses a time-contrastive *quadruplet* loss. We use a quadruplet loss so that the encoder learns that states are the most similar when they occur close in time *and* are associated with the same action. Thus, this encoder also incorporates functional and temporal similarity in states' representations.

---

[2]We have a little abuse of terminology here as $\mathcal{S}$ is different from the traditional state space of a POMDP.

For a given state $s_t$, which we call an anchor state, a positive (visually similar) image $s_t^+$ is one that is at most $\delta$ away in time in an episode and receives the same action from $\pi^*$, i.e., $\pi^*(s_t) = \pi^*(s_t^+)$. We construct two types of negative (dissimilar) states. We find one negative state $s_t^-$ among states that are within the same temporal neighborhood but receive a different action than the anchor. We find a second negative state $s_t^{--}$ among states that are outside the temporal window. Since $s_t^{--}$ is far away from $s_t$ in time, it is highly likely they look dissimilar. Thus, a quadruplet $(s_t, s_t^+, s_t^-, s_t^{--}) \in \mathcal{S}^4$ is constructed. See Figure 3 for an illustration of the idea in Super Mario.

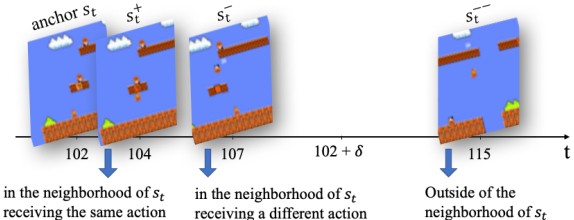

Figure 3: The model trains itself by trying to answer the following questions simultaneously: What is common between the $s_t$ and $s_t^+$ frames? What is different between the similar-looking $s_t$ and $s_t^-$? What is considered visually dissimilar between $s_t$ and $s_t^{--}$?

The quadruplet loss is then defined onm $(s_t, s_t^+, s_t^-, s_t^{--})$ as

$$\mathcal{L}_{\text{quadruplet}}(s_t, s_t^+, s_t^-, s_t^{--}) = \max(\left\| f_\theta(s_t) - f_\theta(s_t^+) \right\|_2^2 - \left\| f_\theta(s_t) - f_\theta(s_t^-) \right\|_2^2 + m_1, 0) \quad (1)$$
$$+ \max(\left\| f_\theta(s_t) - f_\theta(s_t^+) \right\|_2^2 - \left\| f_\theta(s_t^-) - f_\theta(s_t^{--}) \right\|_2^2 + m_2, 0)$$

$m_1$ and $m_2$ are referred to as margins. $m_1$ is used so that, by minimizing $\mathcal{L}_{\text{quadruplet}}$, $f_\theta$ views anchors as at least $m_1$ closer to positives than to the first type of negatives. $m_2$ is used to ensure that the two types of negatives are pushed at least $m_2$ away from each other. The full loss function for the siamese VAE is the quadruplet loss plus the conventional VAE loss function. Pseudocode and further details for our pre-training algorithm can be found in the supplementary material. Once $f_\theta$ is trained, it will be frozen for the down-stream training, such that the property discussed above is preserved.

**Output of the Encoder** The encoder is able to map visually similar states close together in the latent space. In general, we find that the most similar states to a given state are those that almost look identical. The less similar states are generally those that could occur in a small window of time before or after the given state. We show the efficacy of the encoder by aggregating the most similar 30 states together, i.e., the 30 states with the smallest $||f_\theta(x) - f_\theta(x')||_2$. See Figure 4 for an example that shows what states the pre-trained encoder views as similar in Super Mario Bros. level 1-1. This example shows that the states similar to the input are those where Mario is in the process of jumping over a hole in the ground.

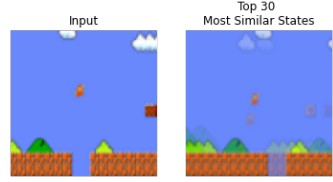

Figure 4: An input state $x$ and top 30 states most similar to $f_\theta(x)$ overlaid.

**Learning Scenario Similarity** Note that the training of $f_\theta$ does not involve the downstream tasks, thus the feature representations, while respecting the visual similarity, may not consider the task relevance, i.e., the scenario similarity, which is task-specific and agent-specific. Similar failure modes in contrastive learning are explored in [42]. Therefore, to overcome the possibility of the $f_\theta$ failing to be useful to the downstream task, we add a linear layer $A : \mathbb{R}^{C' \times H' \times W'} \to \mathbb{R}^{C'H'W'}$ after the encoder. Since $A$ is learned during training, it will further adapt the representations of the input states to the downstream RL task. However, we also want to make sure this linear transformation of $A$ does not undo ProtoX's ability to understand visual similarity. Thus, $A$ needs to be nearly isometric, such that, via[43], if $\left\| A^\top A - I \right\|_2^2 \le \delta$ (where

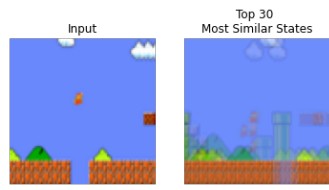

Figure 5: An input state $x$ and top 30 states most similar to $Af_\theta(x)$ overlaid. ProtoX identifies the green pipe being similar to the hole, since they both activate the action of JUMP.

$\delta > 0$), then for all data $x, x'$, $\|Af_\theta(x) - Af_\theta(x')\|_2^2 \le (1+\delta)\|f_\theta(x) - f_\theta(x')\|_2^2$. Therefore, we will include a penalty term in the objective such that the isometry layer allows some evolution of the encoder $Af_\theta$ while still retaining the way $f_\theta$ learned to separate dissimilar states during pre-training.

$$Iso \quad = \quad \left\| A^\top A - I \right\|_2^2, \quad (2)$$

where $I$ is the identity mapping of the appropriate dimension.

The t-SNE plots in Figure 2 explain why the isometry layer benefits the downstream task. We show the 30 states most similar to the same input in Figure 4, this time after the isometry layer, i.e., the 30 states with the smallest $||Af_\theta(x) - Af_\theta(x')||_2$. In contrast to Figure 4, in the example in Figure 5, ProtoX identifies scenario similarity as it considers jumping over a hole is similar to jumping over a green pipe. For more analysis of the isometry layer, please see the supplementary material.

## 3.2 Learning Prototypes

We then add a prototype layer after the isometry layer that automatically learns $K$ prototypes for each action in $\mathcal{A}$, represented by $\{p_k^a\}_{k=1,\cdots,K}^{a \in \mathcal{A}}$. Note that the prototypes are learned via end-to-end training. Each prototype has a shape of $1 \times C'H'W'$, which is the latent representation of an actual image from training data. Given an input state $x$, ProtoX computes the distances $||Af_\theta(x) - p_k^a||_2$ for $k = 1, \ldots, K$ and $a \in \mathcal{A}$. It uses these distances to compute similarity scores between the image and prototypes, following the transformation used in [44]:

$$\text{sim}(Af_\theta(x), p_k^a) = \exp(-\beta||Af_\theta(x) - p_k^a||_2) \tag{3}$$

where $\beta$ controls the concentration of similarity scores and is set to 0.05 in all our experiments.

The similarities are then fed into a fully connected layer with a weight matrix $W \in \mathbb{R}^{|\mathcal{A}|K \times |\mathcal{A}|}$ with $W = (w_{a',k}^a)_{k=1,\ldots,K,a' \in \mathcal{A}}^{a \in \mathcal{A}}$. The subscript represents the prototypes and the superscript represents the output actions, and there are $|\mathcal{A}|K$ prototypes in total. The linear layer outputs $|\mathcal{A}|$ evidence scores, expressed as a weighted sum of similarity score contributions from all prototypes:

$$e^a(x, W) = \sum_{k=1,\ldots,K,a' \in \mathcal{A}} w_{a',k}^a \cdot \text{Sim}(Af_\theta(x), p_k^{a'}), a \in \mathcal{A} \tag{4}$$

The linear model is chosen for its ease of explanation. It could be replaced by other interpretable models like decision trees, following the idea in [9]. The evidence score is computed for every action and the final prediction (action) is made as $\text{argmax}_{a \in \mathcal{A}} e^a(x, W)$. Given this decision rule, we next design an objective function that encourages the model to associate scenarios with an appropriate action to perform.

## 3.3 Objective Function

Since $\pi_\phi$ is a post-hoc explainer for an expert $\pi^*$, we consider two desiderata when training $\pi_\phi$: 1) **fidelity** of imitation, and 2) **interpretability** of the prototypes. We construct the following terms in an objective function for the two purposes.

**Fidelity**   A cross-entropy term, denoted as $\frac{1}{n}\sum_{i=1}^n \mathcal{L}_{CE}(y_i, (e^a(x_i, W))_{a \in \mathcal{A}})$, is included to make sure ProtoX can reproduce the actions of the black-box expert as accurately as possible, which reflects the fidelity in the explanation. Here, $e^a$ is defined in (4).

**Separation and Clustering Terms**   The terms Clst (Clustering) and Sep (Separation) are taken from the ProtoPNet [8]. The Clst term encourages prototypes to be close to the encodings of training instances that have the same action label as the prototypes. The Sep term encourages prototypes to be far from encoded training instances with different action labels as the prototypes. This term enforces consistency with our time-contrastive loss, since both states that image representations should be dissimilar if their corresponding actions differ. The Clst and Sep terms together encourage the model to learn prototype scenarios such that similar scenarios have an associated appropriate action and dissimilar prototype scenarios have different appropriate actions.

$$Sep = -\frac{1}{n}\sum_{i=1}^n \min_{k,a \text{ s.t. } a \neq y_i} ||Af(x_i) - p_k^a||_2, \quad Clst = \frac{1}{n}\sum_{i=1}^n \min_k ||Af(x_i) - p_k^{y_i}||_2, \quad (5)$$

$y_i$ represents the action for instance $i$ and $a$ represents an action that is not $y_i$. Thus, equation (5) means the distance between the encoding of $x_i$ should be far away from prototypes with a different action, which is represented by the separation term. On the other hand, the encoding of $x_i$ should be close to at least one prototype for action $y_i$.

**Representability term** The Rep (representability) term encourages prototypes to be similar to *some* encoded training instances, to ensure the representability of the prototypes. Because our pre-training routine already ensures that encodings $f_\theta(s_t)$ are meaningful representations, we use this Rep term to further ensure that the prototypes are representative of the data. By continuously ensuring that prototypes are close to actual encodings of training data, we justify the success of prototype projection steps that will be explained later. This term is inspired by the Evidence term of [44].

$$Rep = \sum_{k=1}^{K} \sum_{a \in \mathcal{A}} \min_i \|Af(x_i) - p_k^a\|_2^2, \tag{6}$$

**Isometry Term** As discussed previously, the isometry term in formula (2) allows some evolution of the encoder $Af_\theta$ while still retaining the way $f_\theta$ learned to separate dissimilar states.

Therefore, the objective function is

$$\mathcal{L}(\{p_k^a\}_{k=1,\dots,K}^{a \in \mathcal{A}}, A, W) = \frac{1}{n} \sum_{i=1}^{n} \mathcal{L}_{CE}(y_i, (e^a(x_i, W))_{a \in \mathcal{A}}) + \lambda_1 Sep + \lambda_2 Clst + \lambda_3 Rep + \lambda_4 Iso, \tag{7}$$

where $\lambda_i \geq 0$, $i = 1, \dots, 4$, are weighting parameters subject to tuning. The first term encourages high fidelity, and the last four terms are for prototype shaping.

### 3.4 Downstream Training Routine

We train our model with the standard behavior cloning (BC) algorithm, but with the aforementioned surrogate loss $\mathcal{L}$. This training routine is inspired by that of the related ProtoPNet[8].

**Prototype Training and Projection** A prototype layer in ProtoX consists of $K$ tensors of trainable parameters. Let's use one prototype tensor as an example, denoted by $\mathbf{p}$. Since $\mathbf{p}$ is trainable, as the model optimizes the objective in formula (7) via back-propagation, $\mathbf{p}$ is automatically learned. However, one cannot directly obtain the prototypical game states from $\mathbf{p}$ since $\mathbf{p}$ is only a *representation* in the embedding space. One needs to find out which actual state $\mathbf{p}$ corresponds to in the real game state space, represented by pixel values. However, $\mathbf{p}$ may not be able to map back onto an actual game state. Therefore, we do prototype projection, similar to the projection steps in ProtoPNet [8]: every 25 epochs, we map all states in the training data into the embedding space via the encoder $f_\theta$ and isometry layer $A$, then find the one that is closest to $\mathbf{p}$, i.e., $x = \arg\min_{x \in \mathcal{D}} \|Af_\theta(x) - \mathbf{p}\|_2$. This $x$ becomes the prototype, we update $\mathbf{p}$ accordingly by setting $p$ to $Af_\theta(x)$.

**Prototype Merging** Once a ProtoX model is trained and we project prototypes into the space of encodings of real states, we often find that there is a significant amount of repetition among the prototypes. To eliminate the redundant prototypes, we create a new prototype layer consisting only of the unique prototypes. We then also create a new fully connected layer whose weights capture the redundancy present in the previous prototype layer. Supposing that prototype $i = i_1$ is the same as prototypes $i_2, \dots, i_k$ after projection, the connection between prototype $i$ and action $l$ in the fully connected layer is $w_{il} = \sum_{j=1}^{l} w_{i_j l}$. We find that this prototype merging procedure effectively reduced the number of prototypes while retaining the performance of the network. This also means when we build ProtoX, we can over-parameterize it by allowing it to learn many prototypes, but they will eventually be projected into a smaller set of unique prototypes.

## 4 Experiments

We evaluate ProtoX on four Atari/NES game environments from OpenAI Gym and compare the results with two black-box and one interpretable baseline. We first simulate a near-optimal black-box agent and then use different methods to imitate the black-box. We then evaluate the **fidelity** of different methods, measured as the accuracy of their predictions compared to the agent. In addition, we evaluate the sensitivity of ProtoX on flip points, to test whether ProtoX can capture important time steps in a reinforcement learning task. Then we investigate the interpretability of the explanations generated by ProtoX. Finally, we simulate a bad agent and use ProtoX to diagnose its bad behaviors.

### 4.1 Experimental Setup

We use four video-game tasks from OpenAI Gym, namely, Pong, Seaquest and two levels from Super Mario Bros[45]. For each game, we generate a near-perfect black-box agent, denoted as $\pi^*$. Pong

and Seaquest use PPO[46] models from the stable-baselines3[47] package. The experts for Super Mario are also PPO models from [48]. Then we train ProtoX and other methods to imitate each agent. See the Appendix for hyperparameter settings and further details on our experimental design.

We compare ProtoX with three baselines. 1) VIPER[7], which builds a decision tree model to output policies. Since VIPER only works with structured data and does not directly work with images, we extract features from the Pong and Seaquest games. Due to the complexity of Super Mario, we do not benchmark VIPER on the Super Mario games. 2) GAIfO [49], which also learns from observational data like ProtoX but is a black-box model. Unlike ProtoX, GAIfO still interacts with the environment. The goal here is to compare ProtoX's fidelity to another imitation learning method with similar (slightly more) access to the environment. 3) ResNet-BC, which follows the same training procedure as ProtoX except using a ResNet18 model to work with the game frames. ResNet-BC is a black-box model that does not explain an agent. We run ResNet-BC to evaluate how much fidelity is lost due to using prototypes; Both ProtoX and ResNet-BC are trained with the behavior cloning algorithm using 30,000 state-action pairs obtained via an expert trained with PPO.

### 4.2 Fidelity Evaluation

We first evaluate the fidelity of each method, i.e., how often each method agrees with the PPO agent $\pi^*$. For each game, we let the agent generate a test set of $10,000$ state-action pairs $\mathcal{D}_{\text{test}} = \{s_i, \pi^*(s_i)\}_i$. We then apply each method to the states and record their predicted actions $\{\pi(s_i)\}_i$, which is compared against $\{\pi^*(s_i)\}_i$ to compute the accuracy, i.e., $\frac{\sum \mathbb{1}(\pi^*(s_i)=\pi(s_i))}{|\mathcal{D}_{\text{test}}|}$ recorded in Table 1. ProtoX achieved much bet-

Table 1: Fidelity evaluation of different methods.

| **Datasets** | Pong | Seaquest | Mario 1-1 | Mario 8-3 |
|---|---|---|---|---|
| ProtoX | 95% | 96% | 93% | 95% |
| VIPER | 45% | 31% | – | – |
| GAIfO | 19% | 11% | 33% | 25% |
| ResNet-BC | 95% | 98% | 99% | 99% |

ter fidelity than the existing interpretable method, VIPER, since VIPER is a decision tree-based model and does not allow image state inputs. We set the maximum tree depth to 10 for both games, which will produce a tree with thousands of nodes for both games. Increasing the maximum tree depth will slightly increase the fidelity by less than 3 percent, but it will significantly increase the number of nodes in the tree. Yet VIPER still performs much worse than ProtoX. In addition, as numerical features (coordinates of a ball, or a submarine, etc) are more difficult to understand than images in video games. GAIfO also performs poorly. GAIL-based methods are typically only applied to low dimensional problems like joint dynamics [35, 49]. When applied to high-dimensional Atari problems, GAIL has been shown to achieve poor performance [50]. We arrived at a similar conclusion with our experiments.

The comparison of ProtoX and ResNet-BC, even though ResNet-BC is not interpretable, shows that the interpretability design in ProtoX has small negative impact on fidelity, a 3% drop on average. When training ProtoX, we purposely over-parameterize the network by using more prototypes than should be necessary. For example, when training ProtoX for Super Mario Bros. World 1-1, we use 350 prototypes initially. However, after the prototype projection and merging step, we are able to significantly reduce the number of prototypes in a trained ProtoX model, thus making each explanation low in complexity. In the case of Super Mario Bros, we can reduce

Table 2: The number of prototypes in ProtoX models

| **Datasets** | **# of prototypes** |
|---|---|
| Pong | 33 |
| Seaquest | 42 |
| SM 1-1 | 17 |
| SM 8-3 | 12 |

the number of prototypes from 350 to only 17 while retaining relatively high fidelity. We report the number of prototypes for ProtoX models in Table 2. The small number of prototypes ensures that the explanations are easy to follow.

### 4.3 Sensitivity to Flip Points

For an RL agent, usually, the most important explanations are for time steps where the action changes from the previous action. This is because there's often no significant pixel change in two consecutive game frames, and thus a change in the action would be worth investigating. Therefore, if a state receives a different action from its previous state, we call it a *flip point*. The set of flip points can be represented as $\mathcal{F} = \{s_t \in \mathcal{S} | \pi^*(s_t) \neq \pi^*(s_{t-1})\}$. We show an example of a flip

point in Figure 6. At time step $t-1$, the agent presses RIGHT + SPEED to run to the right. In the next time step, the agent presses RIGHT + JUMP to begin jumping over the green pipe.

We evaluate the sensitivity of ProtoX to flip points, i.e., *when the black-box agent changes its action, can ProtoX catch up?* We define sensitivity as the percentage of successful flips, i.e., $\text{sensitivity}(\pi_\phi) = \frac{\sum_{\mathbf{x} \in \mathcal{F}} \mathbb{1}(\pi^*(\mathbf{x}) = \pi_\phi(\mathbf{x}))}{|\mathcal{F}|}$. We collect a total of 10,000 flip points from the test set and report the fidelity of ProtoX on this set in Table 3. This shows that even though flip points are visually similar, ProtoX realizes that they differ and is able to reproduce the agent's action.

Table 3: Sensitivity analysis

| Dataset | ProtoX | VIPER | GAIfO | ResNet-BC |
|---------|--------|-------|-------|-----------|
| Pong | 84% | 44% | 22% | 80% |
| Seaquest | 87% | 24% | 6% | 99% |
| SM 1-1 | 94% | – | 18% | 97% |
| SM 8-3 | 89% | – | 16% | 99% |

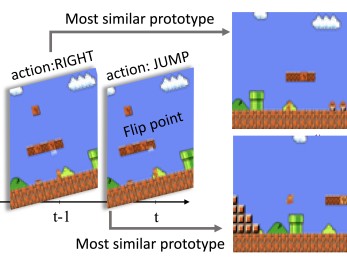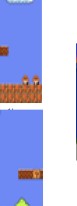

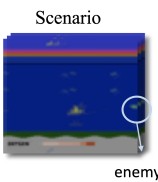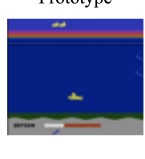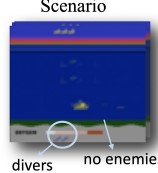

(a) A prototype for action FIRE          (b) A prototype for action UP

Figure 6: A flip point and the corresponding prototype. ProtoX explains the running action in time step $t-1$ with a prototype of Mario running to the right, unimpeded by any obstacle. In contrast, ProtoX recognizes the jumping action in time step $t$ as being similar to another scenario where the expert agent jumped over a green pipe.

Figure 7: Prototypes from Seaquest and their interpretation. The first prototype is for the action "FIRE", which means the submarine tries to shoot enemies to gain points. The prototype represents a situation where the player has shot a missile to the right at an enemy coming in from the right screen edge. The second prototype is for the action "UP", which means the submarine will move up for oxygen. This prototype represents a scenario where the submarine has saved divers, and there are no enemies in the front, so it is safe to go up.

## 4.4   Prototype Interpretation

To investigate the interpretability of ProtoX, we show an example of the prototypes in a ProtoX model learned from Seaquest. The game is an underwater shooter in which the player controls a submarine. Its three goals are 1) to shoot enemies, 2) to rescue divers, and 3) to surface for oxygen when needed.

In Figure 7, we show 2 examples of prototypes. To understand what scenario each prototype represents, we overlay 60 most similar states by plotting the full-color representations of each state with low transparency over one another. Thus, we can faintly see what each scenario represents. For example, for the first prototype in Figure 7, the submarine is located in several places near the bottom center of the screen. But the common feature in this scenario is the enemies to the right of the submarine. The common feature of the second prototype is that the submarine is in the same general position as in the prototype, but with varying air tank levels and a varying number of rescued divers.

**Diagnosing Bad Behaviors**   The explanations provided by ProtoX can also be used to diagnose why an agent makes mistakes. To simulate this use case, we train a bad agent for Super Mario Bros. that never jumps and consequentially is always killed by the first Goomba it encounters. We then train a ProtoX on trajectory data from this agent to see whether ProtoX can explain why the agent fails to jump to avoid the Goomba. Figure 8 (a) shows that Mario sprinted to the right into the Goomba. ProtoX explains this by relating this state to similar prototypes, and we show the most similar prototype in Figure 8 (b), which is a state in the beginning of the game when Mario could sprint to the right because there are no obstacles. To understand *why* the bad agent thinks the input is similar to the prototype, we create an importance map over the bad agent's prototype. The *importance map* is generated by masking the prototype with patches and evaluating the change in the similarity between the masked prototype and the input state. A bigger change indicates higher importance.

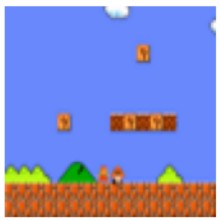 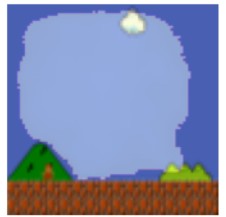 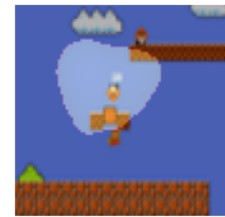

(a) An input state, where Mario runs into a goomba

(b) The most similar prototype from the *bad* agent' ProtoX overlaid by the importance map

(c) The most similar prototype from the *good* agent' ProtoX overlaid by the importance map

Figure 8: For the bad agent, ProtoX views the prototype in (b) as similar, without consideration of any strategic elements. In particular, only the sky seems to matter. In contrast, ProtoX for the good agent selects a different prototype (c) and seems to view the immediate area of Mario hitting the brick to kill the Goomba as important, which explains why the good agent chooses the action "JUMP": he needs to jump to kill the Goomba.

Then, to highlight the most important area, we keep only the top 95% of pixels with the highest values. The rest are set to 0. See the supplementary material for how to generate the importance plot. The importance map is overlaid on the prototype in Figure 8 (b). We deduce from the importance plot that the bad behavior is due to the agent only looks at the blue sky to determine the similarity.

To contrast with the bad agent, we then feed the same input state to another ProtoX trained on a good agent. The new ProtoX identifies a different prototype for the input, shown in Figure 8 (c). In addition, the importance plot shows that the new ProtoX looks at the strategic elements of the games. Specifically, the importance highlights Mario jumping to hit a brick to kill the Goomba. ProtoX thinks this is similar to the input state, where Mario should also jump to kill the Goomba.

# 5    Conclusion

We propose a prototype-based policy explainer, ProtoX, for explaining black-box reinforcement learning agents. ProtoX explains an agent by prototyping its behavior into different scenarios, each represented by a prototype. ProtoX explains an action by comparing the corresponding input with its prototypes, computing the similarity scores between the inputs and prototypes, and then calculating accumulated evidence for each action. ProtoX takes into account both visual similarity and scenario similarity. We use self-supervised training and contrastive learning to learn the notion of visual similarity from expert demonstrations, and allow ProtoX to learn scenario similarity by using an isometry layer during its behavior cloning training. Results show that ProtoX achieved high fidelity compared to VIPER, a black-box model with similar network architecture, and also a GAIfO model whose policy network had a similar architecture.

**Reproducibility**    Our code is available at https://github.com/rrags/ProtoX.

**Discussion of Limitations**    The interpretation of the prototypes requires a basic understanding of the task. For example, to interpret the prototype for UP in Figure 7 users need to know the basic logic for playing the game, and thus can comprehend why the submarine goes up in that scenario. In addition, we have also noticed in the Seaquest game that the expert anticipates the arrival of enemies and shoots at them before they even enter the screen. Thus, ProtoX learns prototypes showing the expert shooting at nothing. ProtoX's mode of interpretation does not inform the users of such anticipatory behavior. One would not be able to understand such prototypes without a basic understanding of the game. Also, ProtoX may learn similar and redundant prototypes. Some pruning techniques could be created to further improve the quality of prototypes.

# 6    Acknowledgements

We thank Nima Safaei for his help with running some baselines in earlier versions of this manuscript. We would also like to thank the anonymous reviewers for their comments and questions, which helped improve this paper. Finally, we would also like to thank Katherine Ragodos for her proof-reading.

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
