# OpenReview forum: "ProtoX: Explaining a Reinforcement Learning Agent via Prototyping"
_NeurIPS.cc/2022/Conference — NeurIPS 2022 Accept_

### Official Review · Reviewer_m9zL · 2022-07-03

**Rating:** 2
**Confidence:** 4
**Soundness:** 1 poor
**Presentation:** 3 good
**Contribution:** 1 poor

**Summary:**

The authors seek to explain black-box deep reinforcement learning agents using prototyping. Prototyping groups together agent behaviors and game-states into prototypical states. They achieve this using ProtoX, which is a self-supervised neural network that uses prototypical states in its forward pass. Prototypes are trained so that they represent groups of similar game states according to visual similarity and action similarity. They're used within ProtoX, which is trained to accurately predict the action taken by the agent given the observation. They demonstrate the method on Pong, Seaquest and two levels from Super Mario Bros. The first show that ProtoX has good fidelity compared with alternative methods (fidelity measure by whether the methods groups states with the same action together). Then they show that ProtoX is sensitive to flip points, which are states that are temporally adjacent but where the agent chooses a different action. Then they demonstrate ProtoX on seaquest and show that prototypes match the scenarios that activate them and show that prototypes of scenarios where the agent exhibits bad behaviours share a similar action to the true action used in the scenario.

**Questions:**

I could be convinced to change my assessment if the authors could show how ProtoX can explain something about the agent's internals, which is necessary to explain why an agent takes an action in a given state. This could perhaps be achieved by augmenting ProtoX to identify feature-level prototypes.

**Limitations:**

The authors have a good discussion of the method's limitations. They are non-obvious and I commend the authors for highlighting them.

**Strengths And Weaknesses:**

The authors do a good job of presenting and explaining their method very clearly. I wasn't previously familiar with prototyping methods and have learned about them thanks to the authors. I am not aware of previous deep RL interpretation methods that take a similar approach; the approach is therefore likely original.

I will judge the paper's quality and significance according to the stated goals of the paper, which are nicely emphasized in the introduction:

(line 23) 'Popular solutions to interpreting a black-box agent include attention-based methods [3] and saliency maps [4, 5, 6]. These explanations, however, only inform users of where the agent “looks at”, but do not provide a more specific explanation for why an agent takes this action in a given state'
(line 30) 'This paper aims to explain a black-box agent that works with unstructured data.'
In summary, the authors state in the introduction that the goal of the paper is an explanation of _why_ an agent takes an action in a given state.

At this point it's worth reflecting on what an explanation of an action (in a given state) actually is, since this is the whole point of ML interpretability work and of this paper. An explanation of an action in a given state should give us an understanding of why the state causes to the action; it should illuminate something about the black-box process that results in the agent taking a particular action in a particular state. Unfortunately, to this end, ProtoX does not succeed. ProtoX only tells us which states are similar to states where an agent took a given action. While prototyping is a valid though (arguably weakly informative) black-box interpretation method, this is only true if the prototypes are on the feature-level, i.e. they should tell us what parts of the state they are prototypes of. ProtoX provides protoypes that aren't on the feature-level; it provides us only with a global picture of the state. It reveals nothing about the black-box process that led from the state to the action. The authors justifiably criticise attention-based methods and saliency maps that tell us little about why an input leads to action; but even those methods tell us markedly more about the internals of the black-box than ProtoX, since those methods at least help us to identify features that the black-box decision making process is using.

Validation of the explanations provided by our interpretability methods is an important step, one that the authors perform in section 4.4 (Diagnosing bad behaviours). But I'm not convinced that this demonstration explains why the agent makes a mistake here (and my reason is in addition to the above criticism that ProtoX doesn't help us to explain the black-box decision making process): ProtoX is trained to group states together that are both visually similar AND that have similar actions. Rather than explain bad behaviour, figure 8 therefore shows only that both the prototype and the real scenario both involve the agent moving right and have similar visual features (because that is what the prototypes were selected to do), but doesn't explain anything about why the bad behaviour happened.

ProtoX is an interesting network, and the ability to group together similar states (in both the visual and action modality) is a praise-worthy feat. But as an approach to help explain _why_ an agent takes an action in a given state (which was the stated goal of the paper) it does not succeed.

Minor criticisms:

The authors write (line 280): "We evaluate ProtoX on four classic control problems from OpenAI Gym and compare the results with two black-box and one interpretable baseline". The environments the authors are referring to are Pong, Seaquest and two levels from Super Mario Bros. I'd encourage the authors not to call these "classic control problems from OpenAI Gym" because gym.envs.classic_control is a different set of environments than Atari games. Their conflation will be confusing to many readers.

(Line 133, footnote). The authors can avoid the abuse of notation by instead describing their RL setup as an MDP: From Wikipedia for POMDP: "an MDP can be reformulated as a POMDP by setting the observation set to be equal to the set of states and defining the observation conditional probabilities to deterministically select the observation that corresponds to the true state."

---

> ### Author Response · Authors · 2022-08-02
> **Response 1 to Reviewer m9zL**
>
> Thank you very much for your comments, Reviewer m9zL! Below we answer your questions and address your comments. We hope you find our proposed solution to your comments satisfactory.
>
> **Q**:  While prototyping is a valid though (arguably weakly informative) black-box interpretation method, this is only true if the prototypes are on the feature-level, i.e. they should tell us what parts of the state they are prototypes of. ProtoX provides prototypes that aren't on the feature-level; it provides us only with a global picture of the state. It reveals nothing about the black-box process that led from the state to the action. The authors justifiably criticise attention-based methods and saliency maps that tell us little about why an input leads to action; but even those methods tell us markedly more about the internals of the black-box than ProtoX, since those methods at least help us to identify features that the black-box decision making process is using.
>
> **A**: Thank you for the suggestion! If prototypes are defined on the feature-level, i.e., as "parts of the game images", users will lose the context of the prototype. For example, for the Pong game, prototypes could be patches containing the balls since the model always needs to look at the ball. For the Mario game, prototypes could be patches containing Mario for the same reason. Without showing the entire state, users will not know where the ball is or where Mario is, or their relative locations to other objects in the game frame. So, we define a prototype as an entire game frame, instead of part of a game frame. However, inspired by your comments that an explanation should tell us what parts of the states the model looks like, we figured out a way to provide explanations on feature-level, specifically, *which part of a prototype $p$ ProtoX looks at when determining the similarity between an input state $s$ and prototype $p$?*
>
> Note that we cannot generate a saliency map over a prototype to answer this question because the prototypes are only $512\times 3\times 3$ in size. So, their gradients would be the same dimension and so too small to upscale to $84\times 84$ and get a meaningful heatmap.
>
> Here's how we generate **importance maps** instead. Given a prototype $p$ which is a latent representation, we first map it back to the image space, to get an actual game state, represented by $x$, with size $84 \times 84 \times 4$, where $84 \times 84$ is the game frame size and 4 refers to 4 consecutive game frames. We then mask $x$ with a *patch*  with size $7 \times 7 \times 4$, which means pixels masked by the patch are all set to 0. This is to remove some part of the game frame from the original state so ProtoX cannot "see" it. And then we feed the masked game state, represented by $x^\prime$ to ProtoX. We compute the similarity between the input state and the masked prototype, i.e., $sim(Af_\theta(s), Af_\theta(x^\prime)$ and evaluate the difference in the similarity compared to the original one, i.e., $\Delta = |sim(Af_\theta(s), p)-sim(Af_\theta(s), Af_\theta(x^\prime)|$. One can think of this as an ablation study: *when certain part of the prototype is removed, does ProtoX still thinks it is similar to the input*? If the similarity difference $\Delta$ is big, it means the area covered by the patch is very critical. We sweep the patch over the entire image, moving the patch horizontally and vertically with a step of 1 pixel each time. Then every pixel (except the ones on the edges) will be evaluated in 49 such evaluations and the 49 $\Delta$s will be averaged, which is the importance of that pixel. Then, to highlight the most important area, we keep only the top 95\% of pixels with the highest values. The rest are set to 0.
>
> We show an example of the importance map in Figure 8 for the prototype identified for the bad agent.  The importance plot suggests that the bad agent did not consider any strategic element of its prototype when computing the similarity of the prototype to the input and it apparently did not even consider the position of Mario: **only the blue sky and a cloud are highlighted**.  It explains why the agent thinks this particular prototype is most similar to the input: the blue skies in the two game states are similar.
>
> To contrast with the bad agent, we then train a good agent with near-perfect performance. we feed the same state to this good agent, who plays action ``JUMP + RIGHT'' for this state. We built a ProtoX to explain this good agent. For this same input, the good ProtoX identifies a different prototype for the input, as shown in Figure 8. We also show the importance map overlayed on the prototype. It shows that this time, strategic elements of the state were considered in the similarity computation. An area containing Mario jumping to hit a brick to kill a zoomba is highlighted, which implies that the good agent thinks this is similar to the input state, that Mario should jump to kill the zoomba.

---

> > ### Author Response · Authors · 2022-08-02
> > **Response 2 to Reviewer m9zL**
> >
> > [*continued*]
> >
> > **Q**: Validation of the explanations provided by our interpretability methods is an important step, one that the authors perform in section 4.4 (Diagnosing bad behaviours). But I'm not convinced that this demonstration explains why the agent makes a mistake here (and my reason is in addition to the above criticism that ProtoX doesn't help us to explain the black-box decision making process): ProtoX is trained to group states together that are both visually similar AND that have similar actions. Rather than explain bad behaviour, figure 8 therefore shows only that both the prototype and the real scenario both involve the agent moving right and have similar visual features (because that is what the prototypes were selected to do), but doesn't explain anything about why the bad behaviour happened.
> >
> > **A**: This question has been addressed in the previous question. Now the importance map shows which part of the prototype is important in being similar to the input state.
> >
> > ---
> >
> > **Q**: The authors write (line 280): "We evaluate ProtoX on four classic control problems from OpenAI Gym and compare the results with two black-box and one interpretable baseline". The environments the authors are referring to are Pong, Seaquest and two levels from Super Mario Bros. I'd encourage the authors not to call these "classic control problems from OpenAI Gym" because gym.envs.classic\_control is a different set of environments than Atari games. Their conflation will be confusing to many readers.
> >
> > **A**: Thank you for noticing this and pointing this out. We have changed "classic control problems from OpenAI Gym" to "Atari game environments from OpenAI Gym" to eliminate any possible confusion.
> >
> > ---
> >
> > **Q**: (Line 133, footnote). The authors can avoid the abuse of notation by instead describing their RL setup as an MDP: From Wikipedia for POMDP: "an MDP can be reformulated as a POMDP by setting the observation set to be equal to the set of states and defining the observation conditional probabilities to deterministically select the observation that corresponds to the true state."
> >
> > **A**: We agree that an MDP can be reformulated as a POMDP, but a POMDP cannot be reformulated as an MDP in general. Our problem is technically a POMDP.
> >
> > ---
> >
> > **Q**: I could be convinced to change my assessment if the authors could show how ProtoX can explain something about the agent's internals, which is necessary to explain why an agent takes an action in a given state. This could perhaps be achieved by augmenting ProtoX to identify feature-level prototypes.
> >
> > **A**: Thank you for being open to accept our improvements. We propose to augment our prototypes with the importance map we created. The new examples provided in the paper (e.g., Fig 8) proved that it could provide some explanation about the agent's internals.  The paper has also been updated. Please see the new section on ``Diagnosing Bad Behaviors" section colored in blue. We hope you find our solution satisfactory.
> >
> > ### Thank you for your suggestions Reviewer m9zL!

---

> > > ### Author Response · Authors · 2022-08-08
> > > **Have the response addressed the reviewer's concerns?**
> > >
> > > Dear Reviewer m9zL,
> > >
> > > Thank you for raising the questions and concerns in the initial review. We have addressed all of them in our response, especially the concerns on the prototypes:  we proposed a way to generate feature-level importance maps to highlight which part of a prototype is important when computing the similarity with an input (see Figure 8 in the updated Diagnosing Bad Behaviors paragraph). The importance maps augment each prototype and provide more insights into an agent's actions. Therefore, we kindly ask the reviewer to reassess the paper in light of our response, and we hope you find our response satisfactory! If there there are more things we can clarify and discuss before the discussion deadline, we are happy to do it too!
> > >
> > > Thanks again for your time reviewing our paper!!

---

> > ### Comment · Reviewer_QULj · 2022-08-09
> > **Increasing score due to additional experiments**
> >
> > The authors have provided significant additional experiments during the rebuttal period, namely additional results for the sensitivity to flip points experiment which I requested, and the new results for generating importance maps that they describe above. I believe these new results significantly strengthen the paper, and as a result I have increased my score.

---

> > > ### Author Response · Authors · 2022-08-09
> > > **Thank you!**
> > >
> > > Thank you reviewer QULj!

---

> ### Comment · Reviewer_XKWc · 2022-08-07
> **Response to R#m9zL by R#XKWc**
>
> Dear R#m9zL,
>
> I'm an RL practitioner and to me, it's much more valuable to have a prototype that shows what the agent thinks is happening in image space than have some similarity in feature space. In feature space, this just causes me more work because I now have to associate this with other inputs' features. I have to go out of my way to craft scenarios (i.e. visual inputs or game states) that I think the policy thinks are currently happening and compare their features to the one of the reference state. I much prefer to have this method give me visual prototypes.
>
> Along these lines, I strongly disagree with your statement that "It reveals nothing about the black-box process that led from the state to the action." because the authors literally show informative examples in figures 4, 5, 6, 7, and 8.
>
> Best,
> R#XKWc

---

> > ### Comment · Reviewer_QULj · 2022-08-08
> > **I concur**
> >
> > I am also an RL practitioner, and I strongly agree with this comment.

---

> > > ### Comment · Reviewer_m9zL · 2022-08-08
> > > **A clarification in response to R#XKWc from R#m9zL**
> > >
> > > I'm not sure it's fully relevant to mention, but since co-reviewers have done so I will mention that I too am an RL practitioner. I add that I'm also an interpretability practitioner.
> > >
> > > > it's much more valuable to have a prototype that shows what the agent thinks is happening in image space than have some similarity in feature space. In feature space, this just causes me more work because I now have to associate this with other inputs' features. I have to go out of my way to craft scenarios (i.e. visual inputs or game states) that I think the policy thinks are currently happening and compare their features to the one of the reference state. I much prefer to have this method give me visual prototypes.
> > >
> > > I think we're using the word 'feature' in a different sense here. When I talk about 'feature space', I'm referring to visual features as distinct from the global image. The original submission provided prototypes that didn't highlight the features in the global image that made the images similar. I was merely pointing out that, unless it's clear what features make the agent consider two images as similar, then it's very hard to tell anything about the agent's internal representations. Since ProtoX of the original submission provided only a set of prototypes without highlighting what was similar about them, it was up to us to guess the features in the image that made the agent think the images were similar. I note that the authors have added extra experiments to address this criticism (to which I give my reponse in the next comment).
> > >
> > > > it's much more valuable to have a prototype that shows what the agent thinks is happening in image space than have some similarity in feature space
> > >
> > > Let's leave aside for the moment the critical point that the prototypes are what the protoX network deems as visually similar, not necessarily what the agent deems visually similar (The only information protoX incorporates about the agent is action-similarity, not visual similarity according to the agent.)
> > >
> > > I want to flag that there are very simple ways of identifying 'a prototype that shows what the agent thinks is happening in image space' that don't involve training an entirely new network with prototypes in order to achieve this. Here is one of them:
> > >
> > > Suppose we record the activations in the agent's policy network from layers 0 to L and the images that caused those activations. Then suppose we cluster the activations in e.g. layer 1. This would let us group together images that the agent considers 'similar' in layer 1. Also suppose we cluster the activations in layer L. This would let us group together images where the agent has a similar action distribution/takes a similar action. Suppose, then, that we combine the activations from different layers (weighting them in some sensible way to normalize them according to vector length). We have thus hypothetically achieved a way to identify prototypical images (i.e. the images associated with each activation cluster) that are both similar in terms of visual- and action-similarity. Note that this is much simpler than ProtoX and tells us as much, if not more, about what is going on inside the agent's black box. It possibly tells us more (should we wish to pursue further experiments with the method) because we are (a) encoding image similarity according to how the agent sees it, not how a separate encoder network sees it and (b) we are looking at the actual activations of the agent rather than merely the input and output.
> > >
> > > Note that this approach is similar to what Mnih et al. (2015) (Extended Data Figure 1) ( https://doi.org/10.1038/nature14236 ) did in one of the very first deep RL papers, although they only examined the activations in the last layer of the policy and looked at nearby activations in a tSNE plot rather than clusters. But the above hypothetical approach and Mnih et al.'s are essentially the same. And they both yield essentially the same outcome as ProtoX, but are much simpler, since they don't involve training a complicated encoder.
> > >
> > > The above mentioned similar methods require access to the agent's inputs, hidden activations, and outputs, while ProtoX requires access only to the agent's inputs and outputs. Given that constraint, an approach like ProtoX is preferable. But it also strikes me as an exceedingly niche application.
> > >
> > > As an intuition pump, a question I pose to both the reviewers and the authors is: In what way does ProtoX improve our understanding of and agent's black box compared with the above-mentioned much simpler methods?

---

> > > > ### Comment · Reviewer_m9zL · 2022-08-08
> > > > **Continued comment: A clarification in response to R#XKWc from R#m9zL**
> > > >
> > > > [Continued]
> > > > > I strongly disagree with your statement that "It reveals nothing about the black-box process that led from the state to the action." because the authors literally show informative examples in figures 4, 5, 6, 7, and 8.
> > > >
> > > > Figure 4 shows an input state x and the top 30 states that are most similar to f_{\theta}(x) overlaid.  It's telling us that the encoder views these images as similar, and since the encoder takes action information into account, it's telling us that these images lead to similar actions. But it's not telling us about the process by which those actions were arrived at by the agent's black box! Similarly for figure 5.
> > > >
> > > > Figure 6 illustrates a 'flip point' where temporally similar images lead to different prototypes. Similarly to figure 4 and 5, this tells us that these images tend to lead to different actions in the agent, but, again, this tells us nothing about what happens inside the agent's black box.
> > > >
> > > > Figure 7, like the previous figures, illustrates prototypes that lead to different actions again without telling us anything about what is going on inside the agent.
> > > >
> > > > Figure 8 in the updated submission uses an importance to illustrate the features that the agent might use in its internal algorithm. The supplementary material that I currently have access to does not describe how they are made. I will assume they're a map made using one of the following mapping methods: a) saliency b) attention or c) perturbation maps on the protoX encoder network. But since protoX is a separate network (a surrogate model), there isn't much/any reason to believe that what is 'important' to the protoX network is what is important to the agent's network. And thus, again, it's doubtful that this method tells us anything about what is actually going on inside the black box of the agent.
> > > >
> > > > But none of the above-mentioned approaches shed much light on the black-box process by which an agent converts an image into an action, which is the stated purpose of ProtoX.

---

> > > > > ### Comment · Reviewer_XKWc · 2022-08-09
> > > > > **Figure Response**
> > > > >
> > > > > > (Fig.4 and 5) it's telling us that these images lead to similar actions
> > > > >
> > > > > Yes. That's what the authors promise and that's what the algorithm delivers and that's all I want.
> > > > > It doesn't help me to understand that neuron #34290 had a high activation and neuron #37194 had a low action but I don't care about that. If you make the assumption that you don't have access to the underlying network or don't want to have access to the underlying agent network, then having `images lead to similar actions` as you said, is incredibly helpful.
> > > > >
> > > > > > (Fig.6) ...again, this tells us nothing about what happens inside the agent's black box
> > > > >
> > > > > And again, this is not the point of the paper. The black box is one because it's inaccessible. The whole premise of the paper and its big appeal is that it treats the agent as input-output system. If you want, you can always take your agent network apart on a case-by-case basis and look at layer-wise attention but that's entirely besides the point.
> > > > >
> > > > > My responses to your comments on Fig.7 and 8 are along the same vein: I'd recommend R#m9zL to reframe the paper and reapproach it from the perspective of the agent being a true black box. Imagine you can't (or won't) access the agent network(s). Reevaluate the paper with this in mind.

---

> > > > > ### Author Response · Authors · 2022-08-09
> > > > > **Response from authors**
> > > > >
> > > > > **We would like to thank all reviewers for their enthusiasm in discussing this paper and the time spent writing reviews. We are sincerely grateful! We hope the effort from all reviewers will help us turn this paper into a work that can provide value to our community and advance the research on interpretability in RL.**
> > > > >
> > > > > Below we address the new comments from reviewer m9zL.
> > > > >
> > > > > ---
> > > > >
> > > > > **Q**: The supplementary material that I currently have access to does not describe how they are made.
> > > > >
> > > > > **A**:
> > > > > How the importance map is generated is described in the response to your initial review.
> > > > >
> > > > > ---
> > > > >
> > > > > **Q**:
> > > > > But since protoX is a separate network (a surrogate model), there isn't much/any reason to believe that what is 'important' to the protoX network is what is important to the agent's network.
> > > > >
> > > > > **A**: Building an interpretable surrogate model to explain a black box is one of the most popular approaches in generating post-hoc explanations, not only in supervised learning, now also in explaining an RL (e.g., VIPER). In fact, the very first paper, LIME [1] is a local linear surrogate model, and then the same authors proposed Anchors [2], a global surrogate model based on rules. The metric, *fidelity*, is originally defined on surrogate models [1, 2, 3, 4, 5], which is how we define our fidelity in this paper. Building a surrogate model is a very popular way to explain a black box when there's no access to its internals, just like in our context.
> > > > >
> > > > > [1] Ribeiro, Marco Tulio, Sameer Singh, and Carlos Guestrin. "" Why should i trust you?" Explaining the predictions of any classifier." Proceedings of the 22nd ACM SIGKDD international conference on knowledge discovery and data mining. 2016.
> > > > >
> > > > > [2] Ribeiro, Marco Tulio, Sameer Singh, and Carlos Guestrin. "Anchors: High-precision model-agnostic explanations." Proceedings of the AAAI conference on artificial intelligence. Vol. 32. No. 1. 2018.
> > > > >
> > > > > [3] Lakkaraju, Himabindu, et al. "Faithful and customizable explanations of black box models." Proceedings of the 2019 AAAI/ACM Conference on AI, Ethics, and Society. 2019.
> > > > >
> > > > > [4] Plumb, Gregory, Denali Molitor, and Ameet S. Talwalkar. "Model agnostic supervised local explanations." Advances in neural information processing systems 31 (2018).
> > > > >
> > > > > [5] Wenbo Guo, Sui Huang, Yunzhe Tao, Xinyu Xing, and Lin Lin.  Explaining deep learningmodels – a bayesian non-parametric approach.  InNeural Information Processing Systems(NeurIPS). 2018

---

> > > > > > ### Author Response · Authors · 2022-08-09
> > > > > > **Response from authors continued**
> > > > > >
> > > > > > [*continued*]
> > > > > >
> > > > > >  **Q**: In what way does ProtoX improve our understanding of and agent's black box compared with the above-mentioned much simpler methods?
> > > > > >
> > > > > > **A**: Thank you for this question.
> > > > > >
> > > > > > First and most importantly, the above-mentioned simple methods **do not work in our setting**: we consider the agent as a black-box. ProtoX is constructed with **no access** to the internals of the agent and the above methods need to access many model details including even the layer-wise outputs. Thus, there is a significant difference in the amount of information that is accessible, which can make one usable in practice and the other not possible to use at all.
> > > > > >
> > > > > > Having said that, we would like to share our thoughts on why ProtoX can explain a black-box agent and address some of the concerns.
> > > > > >
> > > > > > The explanations generated by ProtoX essentially consist of two parts: 1) relating an input to prototypes via encoder and isometry layer to get a feature representation (e.g., similarity vector); and 2) feed this feature representation to an interpretable model (e.g., linear models, which is the fully connected layer) to obtain a prediction.
> > > > > >
> > > > > > First, let us look at the step 2) above, because the existing baseline, VIPER, does almost exactly that: VIPER feeds the state features into an interpretable model (e.g., a decision tree) and train it via supervised learning, using the output from the agent. This is what most post-hoc explainers do: building an interpretable surrogate model to maximize the explanation fidelity.
> > > > > >
> > > > > > Now we discuss step 1). VIPER does not make sense on unstructured data because it directly works with the raw state features, such as the pixel values or encoder outputs, which are not intelligible  to humans. For example, if the input is [0.1, 0.4,1.2, ...], no one would understand what it means. Now let's think about what ProtoX does. One can think of prototype layer in ProtoX as *prototype encoding*, which transforms an unstructured input into a structured vector representation. The output of the prototype  [0.1, 0.4, 1.2,...] now makes sense to us because we know it represents the similarity between an input and a meaningful prototype.
> > > > > >
> > > > > > Reviewer had some concerns on the encoder *The only information protoX incorporates about the agent is action-similarity, not visual similarity according to the agent*. But this is not correct. Even the visual similarity is trained on demonstrations from the expert demonstration. This means, maybe some states are visually similar (mario stands on top of a pipe and mario stands on a brick right above a pipe), but if it never exists in the demonstrations from an agent (because an agent either chooses to jump on a pipe or jump higher on a brick, but never both), this two would not be considered visually similar since they never appear within a short time window. In other words, both scenario similarity (reviewer already agreed) and visual similarity (explained above) are determined by the agent. The selection of the prototypes is combined with step 2) via an end-to-end training to maximize the fidliety with some regularizations. So we conclude that both step 1) and 2) are designed to understand how an agent works.
> > > > > >
> > > > > > Finally, we summarize how ProtoX enables human understanding of an agent's black-box
> > > > > >
> > > > > > - Encoder automatically considers visual and scenario similarity that is specific to the agent.
> > > > > > - Our new importance plot identifies which part of a prototype is considered important
> > > > > > - How the similarities are combined are entirely learned from data, trained via supervised learning, to reproduce the agent's action.
> > > > > >
> > > > > > We hope the clarification can make the contribution of the paper more clear.
> > > > > >
> > > > > > ### Thank you!!

---

> > > > ### Comment · Reviewer_XKWc · 2022-08-09
> > > > **It's a Blackbox**
> > > >
> > > > I think you hit pay dirt here: the main benefit is that it's a black box approach. I don't have to go do network surgery and extract activations at different layers. This can potentially be packaged into a library that you can run off-the-shelf, with absolute disregard for the underlying agent network. To me, that's a huge benefit. This method can work on a million-layer transformer encoder and on a 3-layer MLP, and you as an RL user don't have to carry out any extra steps.

---

### Official Review · Reviewer_XKWc · 2022-07-10

**Rating:** 8
**Confidence:** 4
**Soundness:** 2 fair
**Presentation:** 2 fair
**Contribution:** 4 excellent

**Summary:**

The paper introduces a method for explaining black-box RL agents. This is accomplished by learning "prototype" states and identifying which prototype the current state corresponds to.
Overall a good and potentially impactful paper and I'm recommending acceptance but conditional on some clarification.

**Questions:**

I've included most of my burning questions in the section above. Here's some minor stuff:

- Q.1) In the checklist, you mentioned "the code and data are proprietary" - what data?
- Q.2) You also mentioned all experiments were done with a single seed - did you do a preliminary test to verify that indeed your method isn't sensitive to random seeds?
- Q.3) How sensitive is your method to the loss coefficients?

**Ethics Review Area:**

["I don’t know"]

**Limitations:**

The authors adequately addressed limitations. I'd also add that maybe this method could in the future be combined with visual saliency methods to highlight what part of a prototype is responsible for the agent's decisions.

**Strengths And Weaknesses:**

TL;DR I think the method introduced in this paper is powerful and desperately needed as a tool for the RL community. I just don't have an intuition on how the prototypes are discovered, even after multiple reads. I'm happy to raise my score if the authors address some of my concerns below.

### Strengths

- S.1) Very cool idea and very useful to RL folks.
- S.2) General paper structure good and illustrations helpful.
- S.3) Flip point and fidelity evaluation are good ideas.
- S.4) Easy-to-reproduce Jupyter notebook included.

### Weaknesses

- W.1) I don't fully understand how prototypes are automatically generated, end-to-end. I've seen the different losses in sec. 3.3, I've looked at the pseudocode in the appendix and I had a very quick glance at the Jupyter notebook but I still don't have a good intuition on how they come to be. Maybe other reviewers got behind this more easily. If you could make another attempt at giving me a high-level intuition on the mechanism behind the prototype learning in the rebuttal phase, I'd appreciate that.
- W.2) Analogous to this, there are a few expressions in the paper that are too unclear for me, like lines 61 ("If two states..."), 263 ("During this step..."), 272 ("We then also create...")
- W.3) I think using prototypes for explainability calls for some more qualitative analysis. I'd like to see some more examples with annotations on what they indicate, like Fig. 7. The purpose of the paper is to provide a "policy explainer" and I don't have a feeling that a policy was explained, other than in the "bad behavior" example.
- W.4) Using 30 most similar frames to a given frame and combining them transparently is conceptually a good idea but as a reader, I can't really make out a lot there. I'd recommend either using fewer frames or making a grid of frames or maybe even 5 images of 6 overlayed frames or something similar.
- W.5) 2 minor writing things: (a) I think that figures should stand on their own in a paper and a figure + its caption should contain all the information needed to parse it. In your paper, a lot of times, the caption of the figure is actually in the main text of the paper, not in the caption. For example, the captions for Fig. 6 and 7 are in lines 350+ and 359+ respectively. (b) I like it when at the end of the introduction, there's a list of contributions, either in text or as an actual list but in a very clear and condensed form and highlighted as "contributions".
- W.6) 2 minor experimental things: (a) Why did you choose GAIfO? Why is this a valid baseline? I don't see any similarity or comparability. It's not made to explain anything, it's trained differently, and its networks are different. The only similarity is that it's an imitation-learning method (which is not the focus of your method). (b) In the bad behavior example: you have a policy that always outputs "RIGHT", right? Then if I understand correctly, your ProtoX can't learn nuances between states because all states have the same action. Therefore the conclusion of "it predicts that the danger (goomba) is equal to flat ground" is by definition and a bit pointless, no?

---

> ### Author Response · Authors · 2022-08-02
> **Response 1 to Reviewer XKWc**
>
> Thank you very much for your comments, reviewer XKWc! Below we answer your questions and address some comments.
>
> **Q**: I don't fully understand how prototypes are automatically generated end-to-end. I've seen the different losses in sec. 3.3, I've looked at the pseudocode in the appendix, and I had a very quick glance at the Jupyter notebook but I still don't have a good intuition on how they come to be. Maybe other reviewers got behind this more easily. If you could make another attempt at giving me a high-level intuition on the mechanism behind the prototype learning in the rebuttal phase, I'd appreciate that.
>
> **A**: Sorry for the confusion. Let us try again to see if it's a bit more clear to readers. A prototype layer in ProtoX consists of K $512 \times 3\times 3$ tensors of *trainable parameters*. Let's use one prototype tensor as an example, denoted as $p$.  Since $p$ is trainable, then as the model optimizes the objective in formula (7) via back-propagation, $p$ is automatically learned. However, one cannot directly obtain the prototypical game states from  $p$ since $p$ is only a *representation* in the embedding space. One needs to find out which actual state $p$ corresponds to in the real game state space, represented by pixel values. However, $p$ may not be able to map back onto an actual game state. That is why we do prototype projection. The steps are as follows. Every $n$ epochs, we map all states in the training data into the embedding space via the encoder $f_\theta$ and isometry layer $A$, then find the one that is closes to $p$, i.e., $x=\arg\min_{x \in \mathcal{D}} ||Af_\theta(x) - \mathbf{p}_k||_2.$ This $x$ is the prototypical state, an actual game frame. We then update $p$ accordingly by setting $p$ to $Af_\theta(x)$.
>
> We hope this is more clear to readers.
>
> ---
>
> **Q**: Analogous to this, there are a few expressions in the paper that are too unclear for me, like lines 61 ("If two states..."), 263 ("During this step..."), 272 ("We then also create...")
>
> **A**: Thank you for pointing them out. We have revised those lines. Line 61 is trying to convey that we must balance our goal of including both visual and contextual aspects in our definition of similarity to avoid confusing cases.
> Line 263 summarizes the prototype projection step.  Line 272 refers to re-making the fully connected layer after prototype pruning. Since the number of prototypes is reduced, there will be a mis-match in the number of similarity scores and the number of input nodes to the fully connected layer. Therefore, we have to remove the existing fully connected layer, and make a new one whose input dimension matches the number of prototypes.
>
> ---
>
> **Q**: I think using prototypes for explainability calls for some more qualitative analysis. I'd like to see some more examples with annotations on what they indicate, like Fig. 7. The purpose of the paper is to provide a "policy explainer" and I don't have a feeling that a policy was explained, other than in the "bad behavior" example.
>
> **A**: We have added a prototype identified by a good agent for the same state in Figure 8, to contrast with the bad agent. In addition, following the suggestion of reviewer m9zL, we created importance maps to further explain which part of a prototype makes the agent believe it is similar to an input. This justifies the relation of the input state to the prototype. The important maps will automatically inform users of what makes a prototype similar to an input, providing automatic qualitative analysis.
>
> ---
>
> **Q**: Using 30 most similar frames to a given frame and combining them transparently is conceptually a good idea but as a reader, I can't really make out a lot there. I'd recommend either using fewer frames or making a grid of frames or maybe even 5 images of 6 overlayed frames or something similar.
>
> **A**: Thank you for the suggestion. We actually experimented with aggregating different numbers of most similar states when we wrote the paper. We found that using a few e.g. 5 images to overlay is not enough in our experience since they will all be too similar (e.g. all 5 will be one time step apart with minimal pixel changes). Only when increasing the number to 30 did we find states that do not look identical. We have followed your suggestion to provide a grid of frames in the supplementary material so readers can see them more clearly.

---

> > ### Author Response · Authors · 2022-08-02
> > **Response 2 to Reviewer XKWc**
> >
> > [-*continued*-]
> >
> > **Q**: 2 minor writing things: (a) I think that figures should stand on their own in a paper and a figure + its caption should contain all the information needed to parse it. In your paper, a lot of times, the caption of the figure is actually in the main text of the paper, not in the caption. For example, the captions for Fig. 6 and 7 are in lines 350+ and 359+ respectively. (b) I like it when at the end of the introduction, there's a list of contributions, either in text or as an actual list but in a very clear and condensed form and highlighted as "contributions".
> >
> >
> > **A**: (a) Thanks for this suggestion. We have revised the figure captions to explain each figure such that it isn't necessary to read the body text to understand it. Figures 1, 6, and 7 had their captions replaced with the relevant text from the main body. (b). We have added a list of contributions at the end of the introduction.
> >
> > ---
> >
> > **Q**: Why did you choose GAIfO? Why is this a valid baseline? I don't see any similarity or comparability. It's not made to explain anything, it's trained differently, and its networks are different. The only similarity is that it's an imitation-learning method (which is not the focus of your method).
> >
> > **A**: Thank you for the comment! We had a very hard time finding an appropriate baseline. We didn't find any interpretable baseline except VIPER. So we just wanted to find another black-box model. We chose GAIfO because it is also trained on observational data. But we did papers that explain why GAIfO doesn't work well on video games, which is consistent with our observation. We are considering removing this baseline in the final version and will consider reviewers' suggestions on this.
> >
> > ---
> >
> > **Q**: In the bad behavior example: you have a policy that always outputs "RIGHT", right? Then if I understand correctly, your ProtoX can't learn nuances between states because all states have the same action. Therefore the conclusion of "it predicts that the danger (goomba) is equal to flat ground" is by definition and a bit pointless, no?
> >
> >
> > **A**: ProtoX should understand some nuances between states that are independent of the actions due to the pre-training routine. The time-contrastive pre-training facilitates ProtoX's ability see states as dissimilar even if they have the same action because states outside of the temporal window of anchors must be dissimilar to the anchor. For this particular example you mentioned, we have created an importance map to show where ProtoX ``looks at'' when it computes the similarity between the prototype and the input state. See the new Figure 8 in the paper. The importance map highlights the blue sky and does not look at the ground area, which means ProtoX concludes the prototype is similar from the similarity in the blue sky.
> >
> > ---
> >
> > **Q**: In the checklist, you mentioned "the code and data are proprietary" - what data?
> >
> > **A**: No data are proprietary. We answered no for the code because we didn't plan to include the code and wanted to release the code only after the paper is accepted. But then we changed our mind and included the code anyway. Sorry for the confusion.
> >
> > ---
> >
> > **Q**: You also mentioned all experiments were done with a single seed - did you do a preliminary test to verify that indeed your method isn't sensitive to random seeds?
> >
> > **A**: Yes, we had done a test seeding numpy and PyTorch with the clock time in nanoseconds and got consistent results.
> >
> > ---
> >
> > **Q**: How sensitive is your method to the loss coefficients
> >
> > A: In our experience, the performance of the model in terms of fidelity is not affected much by varying the loss coefficients. However, the prototypes are affected. For example, the Div term has to be high enough or else there will be too many redundant prototypes, which are unhelpful for the sake of explanations. The Clst and Sep terms affect whether or not the association of prototypes to actions is preserved. For example, if they are too low, there will be cases where the top most similar prototype to an input has a different action. We didn't have space to discuss this in the paper, but will include a discussion in the supplementary materials to provide readers some intuition and guideline for parameter tuning.
> >
> >
> > ### Thank you for your comments reviewer XKWc!

---

> > > ### Comment · Reviewer_XKWc · 2022-08-07
> > > **Great response!**
> > >
> > > Dear authors,
> > >
> > > Thanks so much for the detailed response (especially the first point about the intuition on how prototypes are created). I now have a much better sense of how the method works and I'm strongly recommending acceptance since this is well-done and impactful to the community.
> > > I also appreciate the changes you made to the paper. I'd maybe incorporate the first response above into the paper for clarity but that's up to you.
> > >
> > > Best,
> > > R#XKWc

---

> > > > ### Author Response · Authors · 2022-08-07
> > > > **Thank you!**
> > > >
> > > > Reviewer XKWc,
> > > >
> > > > we want to thank you for your feedback and suggestions to the paper, and also your feedback on our response. We are happy the paper was improved and the improvement got acknowledged! We really appreciate it !!!
> > > >
> > > > We have added a new paragraph *Prototype Training and Projection* in Section 3.4 to describe how the prototypes are learned and projected; we have also added in the experiments section why we wanted to compare with GAIfO.
> > > >
> > > > Thank you, Reviewer XKWc!

---

> > > ### Comment · Reviewer_XKWc · 2022-08-07
> > > **regarding GAIfO**
> > >
> > > Regarding GAIfO... while I think it's not a strong baseline, I'd also keep it. I'd just maybe add the justification on why you picked GAIfO to the paper and what you hoped to see there, rather than just "...a GAIL-based method that learns from observational data generated by an agent".

---

### Official Review · Reviewer_QULj · 2022-07-12

**Rating:** 8
**Confidence:** 5
**Soundness:** 3 good
**Presentation:** 3 good
**Contribution:** 3 good

**Summary:**

This paper applies an interpretability method based on discovering prototypical inputs to the problem of interpreting reinforcement learning (RL) policies. The authors create a model that uses a time contrastive loss to encodes states that lead to similar actions and that are similar in time as close to each other in embedding space. They connect this encoding to a behavior cloning model that first applies an isometric transformation to the encoding, then extracts a set of prototypes. The similarity between an input image and the prototypes is computed, and these similarity scores are fed to a fully connected layer that outputs scores for each action. Thus, the model can not only predict which action the agent is likely to take from a given state, but show the most relevant prototype state to explain why it takes that action. Experimental results reveal that the model is highly accurate (although not as accurate as a ResNet + BC baseline), and provides intuitive prototypes that help interpret the behavior of the RL agent.

**Questions:**

Please explain the differences between this work and ProtoPNet [8] and clarify that explanation in the text.

Is the process of merging prototypes manual? This seems time consuming. Is there a way to automate it?

Please add a reasonable baseline to the sensitivity to flip points experiment.

Clarify how time contrastive losses are used in pre-training vs when optimizing Eq. 7.

The caption in Figure 8 refers to the policy running Mario as "he", and makes a statement about what "he thinks", which is anthropomorphizing the RL policy and overclaiming. Please fix.

Since GAIfO performed so poorly, improving its implementation or adding a more competitive baseline would improve the paper.



**Limitations:**

The authors acknowledge the limitations of their method with respect to manually merging prototypes, and the fact that interpreting prototypes requires knowledge of the task.

The societal implications of the work are not discussed. Better interpretability for RL actually could have a number of beneficial societal implications, since it would help with deploying RL to safety-critical human environments (e.g. for robots). This would be good to discuss.

**Strengths And Weaknesses:**

**Significance:** The results of this paper are likely to be significant to the RL community. As an RL researcher, I can confirm that I would find this method useful for interpreting the behavior of a trained agent and debugging why it is failing. Overall the idea for the paper is crisp and compelling.

**Novelty:** The novelty of the paper may be somewhat limited. The authors state that the training procedure is "essentially identical to that of the related ProtoPNet [8]" (line 261). I recommend the authors clarify this (including line 240), since ProtoPNet is about image recognition and therefore it seems unlikely that it uses the same type of time contrastive loss. Nevertheless, the time contrastive loss used here is also similar to (but not identical to) that of Time Contrastive Networks [10]. Essentially the novelty seems to be in the combination and application of these existing techniques to the domain of RL.

**Quality:** The experimental evaluation is relatively thorough, uses fairly complex environments, and compares to reasonable baselines. However, the performance of GAIfO is so poor, that it suggests the authors might need to debug their implementation of GAIfO, or choose a more competitive baseline. The authors should also provide some estimate of variance or error bars for their results in Table 1 and Table 3 (as this is part of the reproducibility checklist).

The "diagnosing bad behaviors" experiment is a compelling illustration of the usefulness of this method.

The sensitivity to flip points experiment was creative and interesting, but there is not much we can conclude from the results unless they include a comparison to some other baseline (such as ResNet-BC). Is ProtoX more sensitive to flip points than other techniques because of its focus on prototypes? If not, this is not a very interesting result.

The overview of related work is well written and complete, and clearly outlines the distinction between this paper and prior work.

**Clarity:** Overall the paper is well written and clear. The examples provided in Figure 1 do a great job of framing the contributions of the paper and making them obvious.

There are a couple of clarity issues however. One is the discussion of the time contrastive (TC) loss as only focusing on "visual similarity". This happens in the intro (lines 60-68), and again in the method (Section 3.1, line 194). Because the TC loss forces states that have similar actions and occur at similar times to be similar, it is not only focused on visual similarity. In fact it incorporates functional and temporal similarity. So referring to it as visual similarity is confusing and misleading. This leads to some awkwardness in the intro, for example where it concludes "Consequently, visually similar states are closer together in the latent space"... the previous sentences have not described why *visually* similar states would be close together at all.

A second clarity issue relates to how the encoder is trained. Section 3.1 and Eq 1 provides an explanation for training the time contrastive encoder. But Section 3.3 appears to use very similar losses, and the differences between how they are applied is not clear. Is the procedure to first pre-train the encoder using the TC losses in Section 3.1, and then continue training the whole network using contrastive losses, but with respect to prototypes, in Section 3.3? Is the encoder frozen after pre-training?

It would be helpful to explain $a$ and $y_i$ in Eq 5.

---

> ### Author Response · Authors · 2022-08-02
> **Response 1 to Reviewer QULj**
>
> Thank you very much for your comments and appreciation of our paper! Below we address your questions.
>
> **Q**: Please explain the differences between this work and  ProtoPNet [8] and clarify the explanation in the text.
>
> **A**: We apologize for the confusing sentence that the training procedure is "essentially identical that of the related ProtoPNet''. What we meant is, we also project the prototypes (which are essential latent representations) to the actual states, like what ProtoPNet did, every $n$ epochs, to make sure the prototypes are meaningful states. We have changed this sentence to "the training procedure is inspired by that of the related ProtoPNet".
>
> There are many differences between the two in terms of the context, the architecture, learning objective, and other model designs.
>
> - ProtoPNet is an image classifier while ProtoX is an explainer.
> - ProtoPNet defines prototypes as patches of an image, such as the wings of a bird, but we define prototypes on the entire game frames, because we need the context to explain an action. If the prototypes are patches, we are likely to get a patch containing a ping-pong ball in the Pong game, without showing the actual location of the balls relative to the paddles. Or we could get patches containing Mario for Super Mario Bros., without knowing where Mario stands.
> - Both the encoder trained with a time contrastive loss and the isometry layer are new in ProtoX, motivated by the reinforcement learning problem properties.
> - The prototype merging is also unique for ProtoX.  We observed the redundancy of prototypes in our problem.
>
> To summarize, ProtoX is not a straight application of ProtoPNet, but rather, borrows the idea of using prototypes for explanation and designs a new model suitable for imitation learning.
>
> ---
>
> **Q**: Is the process of merging prototypes manual? This seems time consuming. Is there a way to automate it?
>
> **A**: The merging process is not manual. First, we wrote a function to identify redundant prototypes. Then, for each set of identical prototypes, the weights are added up like what we described in the paper. The connections between the prototypes and the actions are reconstructed (e.g., there are fewer prototypes and thus fewer connections and fewer weights). All steps are automated.
>
> ---
>
> **Q**: Please add a reasonable baseline to the sensitivity to flip points experiment.
>
> **A**: Thank you for the suggestion! We have added 3 baselines to the sensitivity to flip points experiment. The results are now included in Table 3 in the updated version and copied below. In summary, ProtoX outperforms VIPER and GAIfO, but is not as competitive as ResNet-BC on 3 out of four experiments.
>
> | Dataset | ProtoX | VIPER | GAIfO | ResNet-BC |
> |---------|--------|-------|-------|-----------|
> | Pong    | 84%    | 44%   | 22%   | 80%       |
> | Seaquest | 87%|24%|6%|99% |
> |Super Mario 1-1|94%|-|18%|97%|
> |Super Mario 8-3|89%|-|16%|99%|
>
> ---
>
> **Q**: Clarify how time contrastive losses are used in pre-training vs when optimizing Eq. 7.
>
> **A**: The time contrastive loss is only used in pre-training to construct a good encoder $f_\theta$. This encoder is then used in a ProtoX with its parameters frozen, so it is not affected when optimizing Eq. 7. However, the Iso term in Eq. 7 controls how much ProtoX can change the encodings produced by $f_\theta$ to adapt to the downstream task.
>
> ---
>
> **Q**: The caption in Figure 8 refers to the policy running Mario as "he", and makes a statement about what "he thinks", which is anthropomorphizing the RL policy and overclaiming. Please fix.
>
> **A**: We have fixed the caption in Figure 8. Thank you for pointing it out!
>
> ---
>
> **Q**: Since GAIfO performed so poorly, improving its implementation or adding a more competitive baseline would improve the paper.
>
> **A**: Thank you for the comment! We had a very hard time finding an appropriate baseline. We didn't find any interpretable baseline except VIPER. So we just wanted to find another black-box model. We chose GAIfO because it is also trained on observational data. We tried very hard to tune GAIfO when we wrote the paper, and we tried again during the rebuttal period and did not succeed. We found papers that explain why GAIfO doesn't work well on video games. We are considering removing this baseline in the final version and will consider reviewers' suggestions on this.

---

> > ### Author Response · Authors · 2022-08-02
> > **Response 2 to Reviewer QULj**
> >
> > **Other questions regarding clarity**
> >
> > **Q**: There are a couple of clarity issues however. One is the discussion of the time contrastive (TC) loss as only focusing on "visual similarity". This happens in the intro (lines 60-68), and again in the method (Section 3.1, line 194). Because the TC loss forces states that have similar actions and occur at similar times to be similar, it is not only focused on visual similarity. In fact it incorporates functional and temporal similarity. So referring to it as visual similarity is confusing and misleading. This leads to some awkwardness in the intro, for example where it concludes "Consequently, visually similar states are closer together in the latent space"... the previous sentences have not described why visually similar states would be close together at all.
> >
> > **A**: The motivation for designing the encoder is to achieve visual similarity, so we wanted to emphasize it in the paper. Following your suggestion, have added an explanation in Section 3.1 saying that this encoder also incorporates functional and temporal similarity in states' representations.
> >
> > **Q**: A second clarity issue relates to how the encoder is trained. Section 3.1 and Eq 1 provides an explanation for training the time contrastive encoder. But Section 3.3 appears to use very similar losses, and the differences between how they are applied is not clear. Is the procedure to first pre-train the encoder using the TC losses in Section 3.1, and then continue training the whole network using contrastive losses, but with respect to prototypes, in Section 3.3? Is the encoder frozen after pre-training?
> >
> > **A**: The encoder is obtained by training a VAE whose loss function also includes our time-contrastive quadruplet loss. This is what we've referred to as encoder pre-training even though what's being trained is actually a VAE. We apologize for any confusion this caused. Having completed the training of the VAE, we take the encoder part of the VAE and use it as the encoder of ProtoX. So $f_\theta$ in ProtoX is the encoder part of the VAE. Prior to beginning the training of ProtoX by minimizing Eq. 7, the encoder parameters are frozen. $f_\theta$ is not trainable.
> >
> > So, in summary, there are two distinct training phases that happen consecutively. The first trains a VAE. The second trains ProtoX using the VAE's encoder.
> >
> > **Q**: It would be helpful to explain $y_i$ and $a$ in Eq 5
> >
> > **A**:
> > $y_i$ represents the action for instance $i$ and $a$ represents an action that is not $y_i$. Thus, equation (5) means the distance between the encoding of $x_i$ should be far away from prototypes with a different action, which is represented by the separation term. On the other hand, the encoding of $x_i$ should be close to at least one prototype for action $y_i$.
> >
> > ### Thank you for your comments, reviewer QULj!

---

> > > ### Comment · Reviewer_QULj · 2022-08-03
> > > **Thank you for the detailed response**
> > >
> > > Thank you for providing these new experiments and details, they help clarify my questions significantly. I particularly appreciated the additional results for the sensitivity to flip points experiment.
> > >
> > > Regarding your question about whether you should remove GAIfO entirely, I think it is good to keep it to show you were not able to replicate its results.
> > >
> > > Regarding the other explanations, to the extent you can add these to the paper as well (e.g. with explaining $y_i$ and $a$ in Eq. 5), I think that can improve clarity.

---

### Meta-Review · Area_Chair_HNWu · 2022-08-26

**Recommendation:** Accept
**Confidence:** Less certain

**Metareview:**

This paper proposes ProtoX, a method to identify input prototypes of reinforcement learning agents that are representationally similar to tetst-time inputs. This allows one to surface "relevant training examples" matching the action predictive behavior seen at test time. This is proposed as an "interpretability method to explain agent decisions". Two reviewers have advocated strongly for accepting the paper, while one reviewer has advocated for a strong reject, primarily on the grounds that the paper makes unsubstantiated claims that "ProtoX explains why an agent took a particular action".

After some deliberation with the reviewers, I will recommend accepting the paper. Here is my rationale:

1. there are several tools one can use to better understand how training data and inputs within an example affect predictions. one can take a "formal, axiomatic approach to attribution" (e.g. integrated gradients) or an "informal, non-axiomatic approach to attribution" (e.g. smoothgrad) when it comes to showing "how inputs/training data explain predictions" . Both types of methods have their uses in the toolkit of every deep learning researcher. This paper proposes an *informal* tool to link training data with test time inputs. ProtoX is dataset-level attribution (via computation of prototypes), which is exciting because most explainability techniques focus on example-level attribution. Even though this paper does not prove implementation invariance of ProtoX method, it  as a useful debugging tool for RL practitioners.
2. defining "what does it mean for a neural network to be 'interpretable?" and "why did this neural net make a certain decision" is a broad question within the ML community (bordering on irreducible philosophy), and it's too high of a bar to expect any paper to have a definitive, one-size fits all solution to explainability. I suspect reviewer m9zL found the claims of the paper to imply that they were proposing a more axiomatic attribution method, whereas it is really intended as a tool to diagnose any RL agent.
3. Given that RL agents are very hard to get right, I could see this method (or perhaps an improved version of it with less moving pieces) as a useful tool.

If I were to point out a weakness of the paper, it is that it bears a lot of similarity to non-parametric imitation learning algorithms, e.g.
VINN paper (Surprising Effectiveness of Representation Learning for Visual Imitation Learning, by Pari et al. 2021 https://arxiv.org/pdf/2112.01511.pdf) where k-NN on training set essentially also surfaces attributable examples from a training set wrt test time images, by design. I have no connection to the VINN paper, but would appreciate it if the authors cited this paper or mentioned prior literature on non-parametric methods in learned embedding spaces as an existing tool already used by the RL community.


**Award:**

No

---

### Decision · Program_Chairs · 2022-09-14

Accept